# Arginine Auxotrophy Affects Siderophore Biosynthesis and Attenuates Virulence of *Aspergillus fumigatus*

**DOI:** 10.3390/genes11040423

**Published:** 2020-04-15

**Authors:** Anna-Maria Dietl, Ulrike Binder, Ingo Bauer, Yana Shadkchan, Nir Osherov, Hubertus Haas

**Affiliations:** 1Institute of Molecular Biology, Biocenter, Medical University of Innsbruck, 6020 Innsbruck, Austria; anna-maria.dietl@i-med.ac.at (A.-M.D.); ingo.bauer@i-med.ac.at (I.B.); 2Institute of Hygiene & Medical Microbiology, Medical University of Innsbruck, 6020 Innsbruck, Austria; ulrike.binder@i-med.ac.at; 3Department of Clinical Microbiology and Immunology, Sackler School of Medicine Ramat-Aviv, 69978 Tel-Aviv, Israel; yanka@post.tau.ac.il (Y.S.); nosherov@tauex.tau.ac.il (N.O.)

**Keywords:** *Aspergillus fumigatus*, virulence, arginine, ornithine, siderophores

## Abstract

*Aspergillus fumigatus* is an opportunistic human pathogen mainly infecting immunocompromised patients. The aim of this study was to characterize the role of arginine biosynthesis in virulence of *A. fumigatus* via genetic inactivation of two key arginine biosynthetic enzymes, the bifunctional acetylglutamate synthase/ornithine acetyltransferase (*argJ*/AFUA_5G08120) and the ornithine carbamoyltransferase (*argB*/AFUA_4G07190). Arginine biosynthesis is intimately linked to the biosynthesis of ornithine, a precursor for siderophore production that has previously been shown to be essential for virulence in *A. fumigatus*. ArgJ is of particular interest as it is the only arginine biosynthetic enzyme lacking mammalian homologs. Inactivation of either ArgJ or ArgB resulted in arginine auxotrophy. Lack of ArgJ, which is essential for mitochondrial ornithine biosynthesis, significantly decreased siderophore production during limited arginine supply with glutamine as nitrogen source, but not with arginine as sole nitrogen source. In contrast, siderophore production reached wild-type levels under both growth conditions in ArgB null strains. These data indicate that siderophore biosynthesis is mainly fueled by mitochondrial ornithine production during limited arginine availability, but by cytosolic ornithine production during high arginine availability via cytosolic arginine hydrolysis. Lack of ArgJ or ArgB attenuated virulence of *A. fumigatus* in the insect model *Galleria mellonella* and in murine models for invasive aspergillosis, indicating limited arginine availability in the investigated host niches.

## 1. Introduction

*Aspergillus fumigatus* is the most common mold pathogen of humans, causing a wide range of diseases. In immunocompromised patients, *A. fumigatus* infections can result in life-threatening invasive aspergillosis, with mortality rates reaching 30–90% despite aggressive antifungal treatment [1]. Fungi are eukaryotes and consequently share many of their metabolic pathways with humans, hindering the development of novel fungal-specific drugs. To identify potential targets for antifungal drug development, mutant strains deficient in amino acid biosynthetic pathways not present in higher eukaryotes have been characterized in several fungal pathogens [2]. In *A. fumigatus*, enzymes participating in the biosynthesis of the amino acids tyrosine, phenylalanine, tryptophan, lysine, histidine, cysteine, methionine, and leucine were validated as potential antifungal targets due to virulence attenuation in several infection models [3,4,5,6,7,8]. Reduced virulence caused by arginine auxotrophy was shown for the plant pathogens *Fusarium oxysporum* on melon cultivars [9] and *Magnaporthe oryzae* on rice leaves [10] as well as for *Aspergillus nidulans* in a murine infection model [11]. These data indicate limited arginine availability in different host organisms. 

Arginine is one of the most versatile amino acids involved in multiple cellular processes such as protein biosynthesis, nitrogen metabolism, and urea and nitric oxide (NO) biosynthesis [12,13]. A scheme illustrating arginine and ornithine biosynthesis in *A. fumigatus* based on studies in *Saccharomyces cerevisiae* and *Neurospora crassa* is shown in Figure 1 [14,15]. The first five steps of arginine biosynthesis take place in the mitochondrion. In *A. fumigatus*, the first and fifth steps are catalyzed by acetylglutamate synthase/ornithine acetyltransferase, a bifunctional protein termed ArgJ (ARG7 in *S. cerevisiae*) [16]. The N-acetylglutamate kinase and N-acetyl-gamma-glutamyl-phosphate reductase, which catalyze the second and third steps, are encoded by a single gene *argEF* (ARG5,6 in *S. cerevisiae* and *Candida albicans*) [17,18]. The mitochondrially produced ornithine is either exported into the cytosol by AmcA (ARG11 in *S. cerevisiae*) or converted into citrulline by ornithine carbamoyltransferase termed ArgB (ARG3 in *S. cerevisiae*) [19,20,21]. In the cytoplasm, citrulline is converted to argininosuccinate catalyzed by arginosuccinate synthase (ARG1 in *S. cerevisiae*) and further to arginine catalyzed by argininosuccinate lyase (ARG4 in *S. cerevisiae*). Arginine can be recycled via arginase (AgaA in *A. fumigatus,* CAR1 in *S. cerevisiae*) into ornithine and back to glutamate via proline [14,22]. Cytosolic ornithine is the major precursor for all *A. fumigatus* siderophores, and, consequently, an essential metabolite for adaption to iron starvation [23,24].

In mammals, arginine is classified as a semi-essential or conditionally-essential amino acid, depending on the health status and developmental stage of the individual [25]. In a healthy adult body, L-arginine plasma levels are in the range of 100-200 µM [26]. Not all enzymes required for arginine biosynthesis in mammalian organisms, including humans, are expressed in every tissue [27,28]. Hence, arginine biosynthesis in mammals requires exchange of intermediates between different tissues. Moreover, mitochondrial ornithine biosynthesis displays significant differences between fungal and mammalian pathways. For example, human ornithine biosynthesis within mitochondria does not employ homologs of ArgEF, ArgJ/Arg2 and Arg8 [27,28]. Nevertheless, ArgJ is the only protein of the fungal ornithine/arginine biosynthetic pathway lacking a human homolog as revealed by BLASTP searches (Appendix A). The latter makes ArgJ a potential fungal-specific drug target. Notably, ArgJ has not been functionally analyzed in filamentous fungal species yet.

A previous study showed that inactivation of ArgEF, but not ArgB, led to decreased extra- and intracellular siderophore production and to reduced virulence in the insect model *Galleria mellonella*, underlying the importance of mitochondrial ornithine biosynthesis during iron deficiency in *A. fumigatus* [23]. However, this study suffers from two limitations: the compared ArgEF- and ArgB-lacking mutant strains had a different genetic background (AkuB-lacking derivatives of CEA17 and Af293, respectively) and the mutant strains had not been analyzed in a murine model for invasive aspergillosis. Moreover, the effect of the lack of these genes was analyzed exclusively on glutamine as major nitrogen source.

Here, we analyzed the effects of genetic inactivation of ArgJ, which lacks a human homolog, in comparison to ArgB in the same genetic background (AfS77), including the impact on virulence in the insect model *Galleria mellonella* and two murine pulmonary infection models for invasive aspergillosis.

## 2. Materials and Methods 

### 2.1. Strains, Media, and Growth Conditions

In general, *A. fumigatus* strains were cultured in *Aspergillus* minimal medium (AMM), containing 10 g/L glucose as carbon source and 20 mM *L*-glutamine as nitrogen source with trace elements and salts according to Pontecorvo et al. [29] or in complex medium (CM) containing 10 g/L glucose, 2 g/L peptone, 1 g/L casamino acids, 1 g/L yeast extract, trace elements, and salts, according to Pontecorvo et al. [29]. For preparation of Sabouraud medium, Sabouraud-2% dextrose broth from Merck KGaA, containing 5 g/L peptone from meat, 5 g/L peptone from casein, and 20 g/L glucose, was used. The blood agar medium was prepared with 0.5% sodium chloride and 10% blood. Oatmeal medium was prepared with 60 g/L rolled oats, cooked in water for 30 min and filtered through a cloth before adding 15 g/L agar and autoclaving. Supplements are indicated in the respective experiments. Iron was omitted for iron-starvation conditions (-Fe) and 30 µM FeSO_4_ was added for iron-replete conditions (+Fe). Liquid cultures were inoculated with 1 × 10^6^ conidia/mL medium and incubated for 24 h at 37 °C. Then, 1 × 10^4^ conidia were point-inoculated for plate assays and incubated for 48 h at 37 °C. Conidia for pulmonary mouse infection were cultivated on solid YAG medium (5 g/L yeast extract, 10 g/L glucose, 10 mM MgCl_2_, trace elements, and vitamin solution). 

### 2.2. Deletion of argJ (AFUA_5G08120), argB (AFUA_4G07190) and Reconstitution of ∆argJ and ∆argB in *A. fumigatus*

Deletion of *argJ* and *argB* was performed using the split marker technique as previously described [30,31] and demonstrated in the Appendix A. The *A. fumigatus akuA::loxP* derivative of ATCC46645, AfS77, termed wild type (wt) here [32], was used for genetic manipulation of *A. fumigatus.* Transformants were selected with 0.1 mg∙mL^−1^ hygromycin B (Sigma-Aldrich, Vienna, Austria) on minimal medium plates including 1 M sucrose for osmotic stabilization. For reconstitution of ∆*argJ* and ∆*argB*, functional versions of *argJ* and *argB*, subcloned into the pGEM-T Easy (Promega, Walldorf, Germany) vectors, resulting in plasmids *pargJ* and *pargB*, respectively, were used. Taking advantage of arginine auxotrophy of the ∆*argJ* and ∆*argB* mutants, protoplasts were transformed with *pargJ* and *pargB,* respectively, yielding the complemented strains *argJ^C^* and *argB^C^* (Appendix A). Southern blot analysis confirmed correct genetic manipulations (Appendix A). Genomic DNA was isolated according to Sambrook et al. [33]. Primers used in this study are listed in Appendix A. Fungal strains used in this study are listed in Appendix A. 

### 2.3. Quantification of Biomass and Siderophore Production

For biomass quantification, mycelia from liquid cultures were collected, freeze-dried, and weighed. For siderophore analysis, culture supernatants or lyophilized mycelia were saturated with FeSO_4_ and extracted as previously described [24]. 

### 2.4. Northern Blot Analysis

RNA was isolated using TRI Reagent (Sigma-Aldrich, Vienna, Austria) and peqGOLD Phase Trap (VWR Peqlab, Vienna, Austria) reaction tubes. Ten micrograms of total RNA were analyzed as described previously [34]. Digoxigenin-labeled hybridization probes were amplified by PCR using primers listed in Appendix A.

### 2.5. *Galleria mellonella* Infection Studies

Virulence studies in *G. mellonella* were carried out according to Maurer et al. [35]. Before use, sixth instar larvae, weighing between 0.4 and 0.5 g (SAGIP, Bagnacavallo, Italy), were kept in the dark at 18 °C. Then, 1 × 10^7^ freshly harvested *A. fumigatus* spores were suspended in 20 µL of insect physiological saline (IPS) and injected into the hemocoel via one of the hind pro-legs. Control groups were either injected with 20 µL of IPS or left untouched. Larvae were incubated at 30 °C and survival was monitored for six days [36]. Survival data were evaluated by using Kaplan–Meier survival curves, analyzed with the log-rank (Mantel–Cox) test, utilizing GraphPad Prism 7.00 software. Differences were considered significant at *p*-values ≤0.05. 

### 2.6. Pulmonary Mouse Infection

Two immunocompromised murine models for pulmonary aspergillosis were used: (i) For the non-neutropenic model, six-week-old female ICR mice were immunocompromised by subcutaneous injection with cortisone acetate (300 mg/kg) three days prior to infection, on the day of infection, and three, seven, and eleven days after infection. (ii) For the neutropenic model (CY model), six-week-old female ICR mice were immunocompromised with cyclophosphamide (150 mg/kg in PBS) injected intraperitoneally (Tripathy BS 2010) three days prior and two days post-conidial infection. In addition, three days prior to conidial infection, cortisone acetate (150 mg/kg in PBS) was injected subcutaneously. Fungal strains for infection were grown on YAG medium for three days at 37 °C. Conidia were collected in PBS with 0.2% Tween 20. Mice were infected intranasally with 5 × 10^5^ dormant conidia, suspended in 20 µL of PBS with 0.2% Tween 20 (10 µL in each nostril). Mortality was monitored for 21 days. The statistical differences for mouse survival were calculated using the log-rank (Mantel–Cox) test. Differences were considered significant at *p*-values ≤0.05. This study was carried out in accordance with the recommendations of the Ministry of Health (MOH) Animal Welfare Committee, Israel. The protocol was approved by the MOH Animal Welfare Committee, Israel (protocol number MOH 01-17-035).

## 3. Results

### 3.1. Generation of Arginine Auxotrophic Mutant Strains in *A. fumigatus*

To analyze the role of arginine biosynthesis in *A. fumigatus*, two genes encoding key enzymes of the arginine biosynthetic pathway (*argJ*/AFUA_5G08120 and *argB*/AUA_4G07190) were deleted by replacing the coding region with the hygromycin resistance cassette (*hph*), as described in Material and Methods. As recipient strain, the *A. fumigatus* ATCC46645 derivative AfS77 (∆*akuA::loxP*), which largely lacks non-homologous end joining and is termed wt here, was used [32]. To confirm gene deletion-specific effects, the deletion mutants (termed ∆*argJ* and ∆*argB*) were complemented with a functional *argJ* and *argB* gene copy, respectively (termed *argJ^C^* and *argB^C^*). The inability of ∆*argJ* and ∆*argB* to grow on minimal medium without arginine supplementation enabled re-integration of *argJ* and *argB* at its original locus without an additional selection marker. Correct genetic manipulation was confirmed by Southern blot analysis (Appendix A).

### 3.2. Deletion of argJ and argB Leads to Arginine Auxotrophy in *A. fumigatus*

To analyze the role of *argJ* and *argB* in *A. fumigatus*, growth of ∆*argJ* and ∆*argB* was compared to wt and complemented strains *argJ^C^* and *argB^C^*, respectively, by point inoculating 1 × 10^4^ conidia of the respective strains on minimal, complex, Sabouraud, oatmeal, or blood agar media supplemented with different concentrations of arginine, as shown in Figure 2. Growth analyses demonstrated that lack of either ArgJ or ArgB causes arginine auxotrophy in *A. fumigatus.* On minimal medium, arginine concentrations ≥0.05 mM were required to enable growth of ∆*argJ* and ∆*argB.* Supplementation of 5 mM arginine was needed to fully cure the growth defect, whereby ∆*argB* displayed decreased growth and sporulation compared to ∆*argJ* in the presence of 0.05 and 0.5 mM arginine (Figure 2, AMM). On complex media, which contains yeast extract and peptone as arginine sources, ∆*argB* lacked sporulation in contrast to ∆*argJ.* Supplementation with ≥0.5 mM arginine in complex media rescued the sporulation defect (Figure 2, complex medium). On Sabouraud medium, which contains peptone as arginine source, ∆*argB* displayed a growth and sporulation defect in contrast to ∆*argJ*, which was cured by arginine supplementation (Figure 2, Sabouraud medium). Most remarkably, with oatmeal as sole nutrient source, ∆*argJ* displayed decreased growth, while ∆*argB* failed to grow (Figure 2, oatmeal medium). These results reveal differences in growth behavior between the two arginine-auxotrophic mutants, i.e. the ∆*argB* mutant displayed higher arginine requirement compared to the ∆*argJ* mutant.

Both mutant strains were unable to grow on blood agar, indicating that the arginine content in the blood is too low to support growth (Figure 2, 10% blood agar). The inability of the mutant strains to grow on blood agar is in agreement with the low arginine concentration of about 0.09 mM in human blood [37].

### 3.3. Deficiency of ArgJ, but not ArgB, Decreases Siderophore Production with Glutamine but Not with Arginine as Major Nitrogen Source 

The non-proteinogenic amino acid ornithine is the major precursor of all siderophores and is either produced in the mitochondria from glutamate or in the cytosol from ornithine-derived arginine, as outlined in Figure 1 [14]. Previous studies showed that iron starvation leads to transcriptional upregulation of ornithine/arginine biosynthetic enzymes and increased cellular levels of ornithine and arginine [24]. To investigate the consequences of impaired arginine and ornithine biosynthesis on biomass and siderophore production, ∆*argJ* and ∆*argB* mutant strains, the complemented strains, and the wt strain were cultured in liquid minimal media during iron-limited conditions (-Fe) with glutamine as nitrogen source (20 mM) and 5 mM arginine to support growth of the arginine auxotrophs. Under these conditions, all strains produced similar biomasses (Figure 3A). In contrast to all other strains, however, ∆*argJ* displayed significantly decreased production of both the extracellular siderophore triacetylfusarinine C (TAFC) and the intracellular siderophore ferricrocin (FC) (Figure 3A). As ∆*argJ,* in contrast to all other strains, lacks mitochondrial ornithine synthesis (Figure 1), these data indicate that siderophore biosynthesis is under these conditions fueled mainly by mitochondrial ornithine production.

In a next step, we analyzed biomass and siderophore production during iron starvation in the presence of 20 mM arginine as the sole nitrogen source. This condition largely rescued the siderophore production defect of ∆*argJ* (Figure 3B), indicating that, with arginine as sole nitrogen source, siderophore biosynthesis is fueled mainly by arginase-mediated cytosolic ornithine production.

### 3.4. Arginase is Transcriptionally Repressed by Nitrogen Metabolite Repression in *A. fumigatus*

To analyze transcriptional regulation of marker genes of cytosolic ornithine production (arginase encoding *agaA*), mitochondrial ornithine production (acetylglutamate synthase-encoding *argEF*), mitochondrial ornithine export (*amcA*), and siderophore biosynthesis (ornithine monooxygenase encoding *sidA*), Northern blot analysis was performed (Figure 4). Under iron sufficiency, the transcript level of arginase encoding *agaA* was low with 20 mM glutamine as sole nitrogen source, increased in the presence of 20 mM glutamine plus 5 mM arginine, and was highest with 20 mM arginine as sole arginine source (Figure 4A). Iron starvation increased the *agaA* transcript level compared to iron sufficiency with glutamine as sole nitrogen source and particularly in the presence of 20 mM glutamine plus 5 mM arginine. Similar to *agaA*, iron starvation caused upregulation of *argEF, amcA,* and *sidA* (Figure 4A). However, in contrast to *agaA*, transcript levels of these genes were not affected by arginine availability. 

Taken together, these results confirm that *agaA*, *argEF*, *amcA,* and *sidA* are transcriptionally upregulated during iron starvation compared to iron sufficiency, as reported previously [19,24,38]. Moreover, the data indicate that arginase is transcriptionally activated by arginine and repressed by nitrogen metabolite repression (i.e., repression in the presence of a primary nitrogen source such as glutamine) in *A. fumigatus* during iron sufficiency, as previously shown in *A. nidulans* [39]. Interestingly, nitrogen metabolite repression (20 mM glutamine and 5 mM arginine) was less pronounced during iron starvation. 

Next, we analyzed the expression of *agaA*, *argEF*, *amcA,* and *sidA* in ∆*argJ* and ∆*argB,* in comparison to the control strains. With one exception, both ∆*argJ* and ∆*argB* displayed similar expression patterns with 20 mM glutamine plus 5 mM arginine as well as with 20 mM arginine as nitrogen sources: with 20 mM glutamine plus 5 mM arginine, both mutant strains showed reduced *agaA* transcript levels. This might be explained by the increased arginine requirement due to the arginine auxotrophy. The increased metabolization of arginine in these strains most likely reduces the cellular arginine content, which decreases expression of *agaA* as this gene is induced by arginine (see above). Taken together, inactivation of either ArgB or ArgJ does not impact the expression of *agaA, argEF, amcA,* and *sidA* differently.

### 3.5. Deficiency of ArgJ, but Not ArgB, Impairs Growth during Iron-Limited Conditions with Glutamine as Nitrogen Source, When Reductive Iron Assimilation Is Blocked

To further elucidate the role of ArgJ and ArgB in iron homeostasis of *A. fumigatus*, both arginine-auxotrophic mutants and control strains were grown on iron-poor, solid minimal medium with the ferrous iron-specific chelator bathophenanthroline disulfonate (BPS), which blocks reductive iron assimilation [40]. Blocking reductive iron assimilation with BPS, which forces iron acquisition exclusively via siderophores, severely impaired growth of ∆*argJ* but not of ∆*argB* with 20 mM glutamine plus arginine as nitrogen source (Figure 3C). In contrast, there was no difference in growth between the two mutant strains without glutamine as the sole nitrogen source (Figure 3C). The fact that weaker growth of ∆*argJ* compared to ∆*argB* was seen only in the presence of BPS indicates that reductive iron assimilation compensates for the loss of siderophore-mediated iron acquisition with glutamine as major nitrogen source. It is important to note that the liquid growth assays (Figure 3A,B) were carried out without BPS, explaining similar biomass production of ∆*argJ* and ∆*argB* with glutamine as major nitrogen source. 

### 3.6. Arginine Biosynthesis Is Crucial for Full Virulence of *A. fumigatus* in the Insect Model *Galleria mellonella*

To analyze the role of arginine and ornithine biosynthesis in *A. fumigatus*, we compared wt, ∆*argJ*, ∆*argB*, *argJ^C^*, and *argB^C^* in the *G. mellonella* infection model (Figure 5A,B). Three days after infection, all larvae infected with the wt or the complemented strains *argJ^C^* and *argB^C^* died. ArgJ-deficiency significantly attenuated virulence of *A. fumigatus* compared to the wt and *argJ^C^* (*p* < 0.0001 vs. wt, *p* < 0.0001 vs. *argJ^C^*), i.e., it delayed killing to Day 4. ArgB-deficiency rendered *A. fumigatus* almost avirulent, i.e., infection with ∆*argB* resulted in 80% survival at Day 6—only 10% lower than the survival rate in the untouched control (*p* < 0.0001 vs. wt, *p* < 0.0001 vs. *argB^C^*).

Taken together, these data demonstrate that both ArgB and ArgJ are essential for full virulence of *A. fumigatus* in the insect model, whereby ArgB-deficiency affected virulence more dramatically compared to ArgJ-deficiency. 

### 3.7. Arginine Biosynthesis Is Crucial for Full Virulence of *A. fumigatus* in Murine Models for Invasive Aspergillosis

To analyze the role of arginine biosynthesis in terms of pathogenicity of *A. fumigatus,* two immunocompromised murine models of infection were assessed: (i) a non-neutropenic pulmonary infection model, in which mice were immunosuppressed with cortisone acetate (CA); and (ii) a neutropenic pulmonary infection model in which mice were immunosuppressed with cyclophosphamide (CY) and cortisone acetate. Therefore, ten six-week-old ICR mice per group were intranasally infected with 5 × 10^5^ conidiospores of the fungal strains. The virulence of the complemented strains *argJ^C^* and *argB^C^* was not analyzed in the murine models as they displayed wt-like behavior in all in vitro experiments and the insect virulence model. Mortality was monitored over a period of 21 days (Figure 5C-D). One hundred percent mortality of the wt was monitored within five days in the CA model and eleven days in the CY model post-infection. In contrast, in both models, deletion of *argJ* and *argB* resulted in higher survival curves compared to mice infected with the wt. Both arginine auxotrophs showed statistically significant attenuated virulence in the CA model compared to mice infected with the wt: 70% survival of ∆*argB* (*p* < 0.0001 vs. wt) and 40% survival of ∆*argJ* (*p* < 0.0001 vs. wt), as shown in Figure 5C. In the CY model, both mutant strains exhibited attenuated virulence compared to the wt (Figure 5D), whereas differences were statistically significant only for ∆*argB*: 55% survival of ∆*argB* (*p* = 0.0002 vs. wt) and 35% survival of ∆*argJ* (*p* = 0.1268 vs. wt). In all tested models, virulence attenuation of both arginine auxotrophs was demonstrated, with a stronger effect of *argB* deletion compared to *argJ* deletion. 

These virulence defects of ∆*argJ* and ∆*argB* are supported by the growth defects of ∆*argJ* and ∆*argB* on blood agar (Figure 2). The decreased virulence of ∆*argJ* and ∆*argB* in murine models for invasive aspergillosis indicates that arginine biosynthesis is crucial for adaptation of *A. fumigatus* to the mammalian host niche and that the lack of arginine biosynthesis cannot be fully compensated by the availability of arginine in the host.

## 4. Discussion

Biosynthesis of arginine is found in bacteria, plants, and fungi, while this amino acid is classified as a semi-essential or conditionally-essential amino acid in mammals [25]. A BLAST search revealed that the bifunctional enzyme ArgJ (acetylglutamate synthase/ornithine acetyltransferase) is the only arginine biosynthetic enzyme lacking homologous proteins in humans (Appendix A), making this enzyme a potentially interesting antifungal drug target. Although the ornithine carbamoyltransferase-encoding *argB* gene is used as a selectable transformation marker for genetic engineering of A*spergillus* spp. [41,42], the virulence potential of this gene has never been investigated in a murine pulmonary infection model. Arginine biosynthesis is intimately linked to biosynthesis of ornithine, the precursor of extra- and intracellular siderophores (Figure 1). Previous studies demonstrated, that lack of siderophore production leads to avirulence in both murine and *G. mellonella* infection models [40,43]. Therefore, arginine biosynthesis combined with precursor supply for siderophore biosynthesis might represent a particularly interesting target for antifungal treatment. 

Here, we show that inactivation of ArgJ, the function of which has not been investigated before in *A. fumigatus*, indeed results in arginine auxotrophy similar to inactivation of ArgB (Figure 2). With glutamine as major nitrogen source (20 mM) and limited arginine supplementation (5 mM), inactivation of ArgJ but not of ArgB resulted in decreased siderophore production (Figure 3B). Inactivation of ArgJ but not of ArgB blocks mitochondrial ornithine production, while inactivation of neither ArgJ nor ArgB affects cytosolic ornithine production via arginase (Figure 1). Therefore, these data indicate that siderophore production is mainly fueled by mitochondrial ornithine production with glutamine as major nitrogen source. In contrast, neither ArgJ nor ArgB displayed defective siderophore production with arginine (20 mM) as major nitrogen source (Figure 3), indicating that siderophore biosynthesis is fueled by cytosolic ornithine production via arginase-mediated hydrolysis of arginine under this condition (Figure 1). In line, inactivation of ArgJ but not of ArgB decreased growth during iron starvation with glutamine but not with arginine as major nitrogen source when alternative reductive assimilation was blocked with BPS (Figure 3C). The role of ArgJ and mitochondrial ornithine production and siderophore biosynthesis is in agreement with previous studies analyzing the role of the mitochondrial arginine biosynthetic enzyme ArgEF and the mitochondrial ornithine exporter AmcA [19,23]. Inactivation of ArgEF, which is similar to ArgJ essential for mitochondrial ornithine biosynthesis, was found to decrease the cellular ornithine content with glutamine as major nitrogen source [23]. This explains the decrease in siderophore production as ornithine is a major siderophore precursor. In agreement with the different impact of glutamine versus arginine on siderophore production, arginase has previously been shown to be transcriptionally activated by arginine and repressed by ammonium via nitrogen metabolite repression in *A. nidulans* [39], which holds true also for *A. fumigatus,* as shown here (Figure 4). Moreover, we confirmed that cytosolic ornithine production (arginase encoding *agaA*), mitochondrial ornithine production (acetylglutamate synthase encoding *argEF*), mitochondrial ornithine export (*amcA*) and siderophore biosynthesis (ornithine monooxygenase encoding *sidA*) are transcriptionally upregulated during iron starvation compared to iron sufficiency as reported previously [19,24,38]. In contrast to *agaA*, transcript levels of *argEF*, *amcA,* and *sidA* were not affected by arginine availability. 

The virulence potential of both arginine auxotrophic mutant strains (∆*argJ* and ∆*argB*) was assessed in the insect model *G. mellonella*, in a non-neutropenic (CA), and a neutropenic (CY) murine model for invasive aspergillosis (Figure 5). Both mutant strains displayed attenuated virulence in all three host models. Interestingly, lack of ArgB attenuated virulence more severely compared to lack of ArgJ in the *G. mellonella* and both murine immunosuppression models (Figure 5A–C). This finding is rather surprising as we expected that ArgJ would be more important for virulence compared to ArgB as its lack causes not only a defect in arginine biosynthesis but also siderophore biosynthesis, which has previously been demonstrated to be essential for virulence [23,24]. Nevertheless, these differences in virulence between the two arginine auxotrophic mutants match their growth phenotype because ∆*argB* showed poorer growth compared to ∆*argJ* under conditions of limiting arginine supply, best seen on complete, Sabouraud, and oatmeal media (Figure 2). The differences in growth and virulence phenotype of the two arginine auxotrophic mutants might be explained by effects of cellular accumulation of different intermediates of the arginine biosynthetic pathway in the two mutants (Figure 1). Moreover, in the virulence models, uptake of different intermediates of the arginine biosynthetic pathway from the host might play a role. Previously, inactivation of ArgB was found to result in wt-like virulence, while lack of ArgEF attenuated virulence in the insect model *G. mellonella* [23]. It is important to note that the previous study used different genetic backgrounds compared to the current study: deletion of *argB* was performed in a derivative of *A. fumigatus* Af293 [19,20,21], which might indicate that the genetic background interferes with the effect of arginine auxotrophy on virulence in the *G. mellonella* model. In this context, it is interesting to note that several studies revealed significant differences in physiological responses to abiotic stimuli and virulence in murine models of invasive pulmonary aspergillosis between *A. fumigatus* Af293 and other *A. fumigatus* isolates including commonly used *A. fumigatus* CEA10 [44,45,46,47,48]. Sugui et al. [47] concluded that Af293 possesses a nutritional deficiency.

Taken together, this work confirms previous studies showing that inactivation of ArgB, and also of ArgJ, results in arginine auxotrophy. Similar to lack of ArgEF or AmcA [19,23], inactivation of ArgJ impaired siderophore production and consequently adaptation to iron starvation due to its role in mitochondrial ornithine production in the presence of glutamine as nitrogen source. Here, we additionally demonstrate that the impact of ArgJ on siderophore production is dependent on the nitrogen source, i.e. the lack of ArgJ did not affect siderophore production with arginine as major nitrogen source. Moreover, we analyzed for the first-time the pathogenicity of arginine auxotrophic *A. fumigatus* mutants in murine aspergillosis models, revealing the importance of arginine biosynthesis during infection by *A. fumigatus,* which indicates that the arginine supply in the host niche is too low to support full virulence. 

## Figures and Tables

**Figure 1 genes-11-00423-f001:**
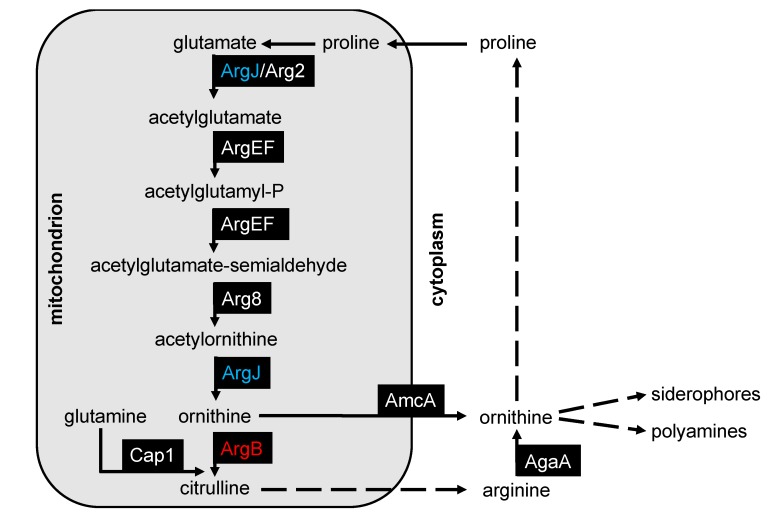
Schematic illustration of arginine/ornithine biosynthesis in *Aspergillus fumigatus*. Starting with glutamate, ornithine is synthesized in the mitochondrion in five enzymatic steps, and either directly transported to the cytoplasm via the ornithine transporter AmcA or converted into citrulline before being transported to the cytoplasm. Citrulline is converted into arginine and further into ornithine via arginase. Ornithine serves as precursor for extra- and intracellular siderophore production. ArgJ (AFUA_5G08120, in blue), bifunctional acetylglutamate synthase/ornithine acetyltransferase; Arg2 (AFUA_2G11490), acetylglutamate synthase; ArgEF (AFUA_6G02910), bifunctional acetylglutamate kinase and N-acetyl-gamma-glutamyl-phosphate reductase; Arg8 (AFUA_2G12470), acetylornithine aminotransferase; ArgB (AFUA_4G07190; in red), ornithine carbamoyltransferase; AmcA (AFUA_8G02760), mitochondrial ornithine exporter; AgaA (AFUA_3G11430), arginase; Cap1 (AFUA_5G06780), carbamoyl-phosphate synthase. Dashed lines indicate metabolic routes involving more than one enzyme. Refer to Appendix A for details regarding homology of the enzymes.

**Figure 2 genes-11-00423-f002:**
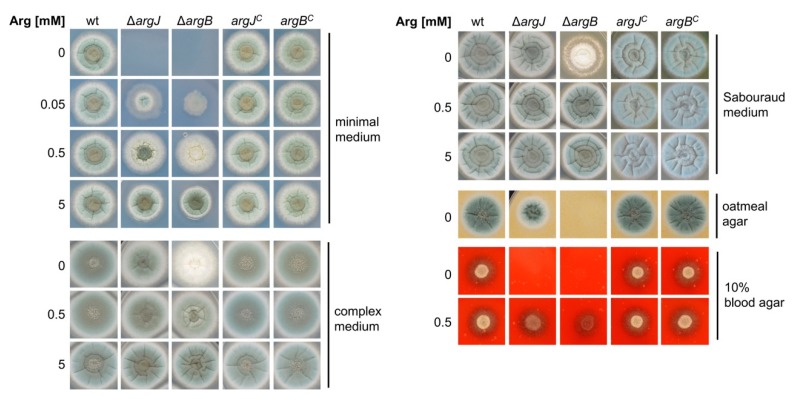
Deletion of *argJ* and *argB* leads to arginine auxotrophy in *A. fumigatus.* Wild-type (wt), mutant (∆*argJ* and ∆*argB*), and complemented (*argJ^C^* and *argB^C^*) strains were point inoculated (1 × 10^4^ conidia) on minimal medium (AMM), complex medium (CM), Sabouraud medium, oatmeal medium, or 10% blood agar containing increasing arginine concentrations and grown at 37 °C for 48 h. Green coloration of the colony reflects sporulation due to green pigmentation of spores.

**Figure 3 genes-11-00423-f003:**
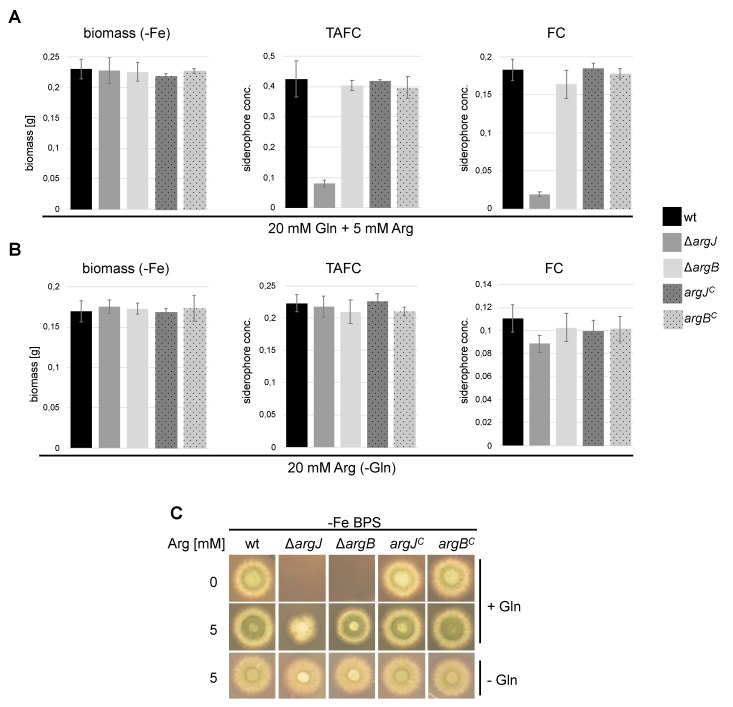
Lack of ArgJ, but not ArgB, impairs adaption to iron starvation with glutamine but not with arginine as major nitrogen source. (**A**) wt, ∆*argJ*, and ∆*argB* were grown for 22 h at 37 °C in liquid minimal medium under iron-depleted conditions (-Fe) in the presence of 5 mM arginine and 20 mM glutamine. ∆*argJ* displayed significantly decreased production of the extracellular (TAFC) and intracellular (FC) siderophores (TAFC; *p* = 0.0002 vs. wt; FC; *p* < 0.0001 vs. wt) while ∆*argB* showed wt-like siderophore production. (**B**) Iron-starved media (-Fe) with 20 mM arginine as sole nitrogen source (-Gln) rescued the siderophore production defect of ∆*argJ* (TAFC, *p* = 0.354 wt vs. ∆*argJ*; FC, *p* = 0.024 wt vs. ∆*argJ*). (S**C**) Wt, ∆*argJ,* and ∆*argB* were point inoculated on solid minimal medium containing 0.2 mM of the iron chelator BPS (-Fe), 5 mM arginine, 20 mM glutamine, or no glutamine (-Gln). Growth was scored after 48 h at 37 °C. In (A,B), mean values of three replicates ± standard deviations are shown.

**Figure 4 genes-11-00423-f004:**
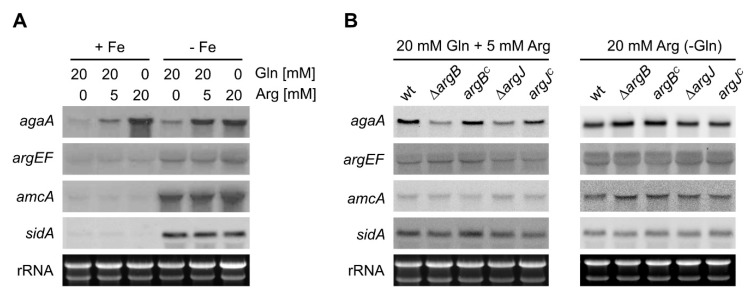
In contrast to *argEF, amcA* and *sidA*, expression of *agaA* is induced by arginine and repressed by nitrogen metabolite repression in wt (**A**); inactivation of either ArgB or ArgJ does not impact the expression of *agaA, argEF, amcA,* and *sidA* differently. Fungal strains were grown for 17 h (**A**) or 22 h (**B**) at 37 °C in liquid minimal medium under iron sufficiency (+Fe) and iron starvation (-Fe) with the indicated nitrogen sources: 20 mM glutamine (Gln), 20 mM arginine (Arg) or 20 mM Gln plus 5 mM Arg. Ethidium bromide-stained ribosomal RNA is shown for control for RNA loading and quality.

**Figure 5 genes-11-00423-f005:**
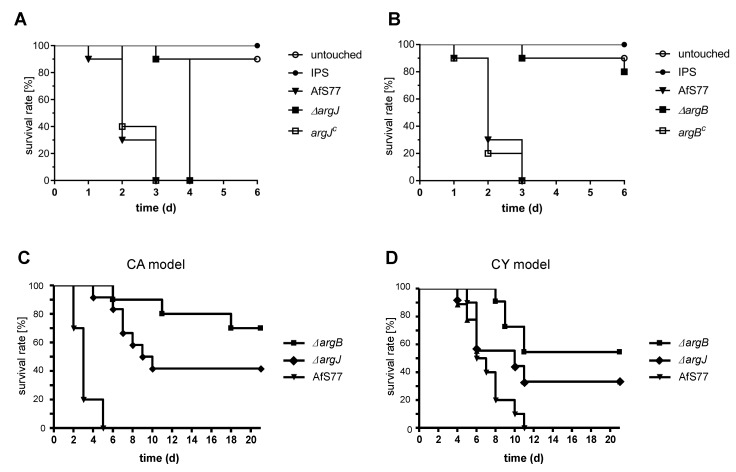
Lack of ArgJ and ArgB attenuates virulence of *A. fumigatus* in the insect model *G. mellonella* and in murine models for invasive aspergillosis. (**A**,**B**) *G. mellonella* larvae (n = 40 larvae per group) were infected with 1 × 10^7^ conidia of the respective strains and survival was monitored for six days. Control cohorts were injected with insect buffered saline (IPS, n = 20) or left untouched (n = 20). Lack of ArgJ (∆*argJ*) or ArgB (∆*argB*) attenuated virulence of *A. fumigatus* in *G. mellonella* compared to the wt and the complemented strains *argJ^C^* and *argB^C^* (∆*argJ*, *p* < 0.0001 vs. wt and *argJ^C^*; ∆*argB*, *p* < 0.0001 vs. wt and *argB^C^*). (**C**) Mice were immunocompromised with cortisone-acetate (CA model, non-neutropenic, n = 10 mice/group) or with (**D**) cyclophosphamide and cortisone acetate (CY model, neutropenic, n = 10 mice/group) and intranasally infected with 5 × 10^5^ conidia of the wt, ∆*argJ,* and ∆*argB*. Survival was monitored for 21 days. In the CA model (**C**), lack of ArgJ (∆*argJ*) and ArgB (∆*argB*) attenuated virulence of *A. fumigatus* compared to the wt strain (∆*argB, p* < 0.0001 vs. wt; ∆*argJ*, *p* < 0.0001 vs. wt). In the CY model (**D**), lack of ArgB, but not ArgJ, attenuated virulence of *A. fumigatus* compared to the wt strain (∆*argB, p* = 0.0002 vs. wt; ∆*argJ*, *p* = 0.1268 vs. wt, not significant).

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
