# Peer review of "Arginine Auxotrophy Affects Siderophore Biosynthesis and Attenuates Virulence of Aspergillus fumigatus"

_genes, 2020, doi:10.3390/genes11040423_

Round 1
Reviewer 1 Report
The authors has answered to my comments, and the manuscript was improved. I think that the manuscript can be accepted for publication in its current form.
Reviewer 2 Report
The authors addressed all issued raised by reviewers and revised previous version (genes-708865). So it can be published in GENES.
Reviewer 3 Report
Comments were taken into consideration and additional experiments were done and presented. English language check (especially articles before nouns) is still required.
This manuscript is a resubmission of an earlier submission. The following is a list of the peer review reports and author responses from that submission.
Round 1
Reviewer 1 Report
In this manuscript, the authors finely demonstrated functional roles of enzymes of arginine biosynthetic pathway in pathogenic fungus Aspergillus fumigatus. As a result, ArgJ and ArgB were shown to play a role in virulence of the fungus. Those experiments are well-done.
I think that the manuscript can be published after some concerns below were addressed by the authors.
L209-210: “glutamine transcriptionally repress arginase, an enzyme that converts arginine into ornithine in the cytosol, via nitrogen metabolite repression.” This should be confirmed by some expression analysis, such as RT-PCR. Alternatively, determination of cytosolic ornithine amount would be a help for understanding what was happen in the cells when grown in the presence of glutamine. There might be possibility that siderophore biosynthetic pathway was transcriptionally repressed under the condition, which led to reduction of TAFC and FC. In Figure 3, panels are wrong. (B) <-> (C) L233-237: The authors insisted that argJ mutant strain impaired the growth under iron-depleted condition in the presence of glutamine. However, when cultured in liquid medium, the biomass of argJ mutant was comparable to that of wt and argB mutant. This seems to be contradicted. Please provide possible explanation for this. In Figure 4: The symbols should be more readable, in particular for Fig. 4A and 4B. L308-309: “in agreement, decreased growth during iron starvation.” Please be careful.
Reviewer 2 Report
In this manuscript, the authors describes the roles of two arginine biosynthesis related genes argB and ArgJ in opportunistic human pathogen Aspergillus fumigatus. The authors generated mutant strains and examined phenotypes and virulence of these mutants. Some results which are related to ArgB have been published elsewhere, but ArgJ data are novel. There it can be published in Genes after revision.
In Figure 3, there were missing results for complemented strains and it should be added. In murine models, complemented strains were missing. Transcript levels of genes involved in biosynthesis of ornithine/arginine can be examined to improve this manuscript. Line 267, there are missing heading number (3.6). Line 158, Protocol number should be provided.Reviewer 3 Report
The current manuscript provides an analysis of two arginine biosynthetic enzymes, ArgJ and ArgB, in the human fungal pathogen A. fumigatus. Assuming that authors will answer questions below, I recommend this manuscript for publication.
Previously, a gene encoding ArgB was deleted and characterized in another strain of A. fumigatus. In the current study, authors used AfS77 as a background strain. Interestingly, when argB was deleted in AfS77, the obtained strain showed attenuated virulence in all infection models, whereas the deletion of the same gene in Af293 background strain did not result in attenuated virulence. Can authors provide a potential explanation for such a difference? Was the same background-specific effect reported for other genes in A. fumigatus?
The strain lacking ArgB showed severe attenuation in virulence in contrast to ∆argJ. Authors explain this but the fact that ∆argB showed poorer growth compared to ∆argJ under conditions of limiting arginine supply as observed only on complete medium. Interestingly, such an effect was not observed on the minimal media with low levels of Arg added. Could other components of the complete media interfere with the result? Was other types of rich media used to analyse difference in growth of the two mutants, like Sabouraud dextrose?
Line 189: How the explanation that the difference in growth of two mutants is observed because ∆argB mutant displayed higher arginine requirement compared ∆argJ, corresponds with the Figure 3B? On that figure the opposite effect can be seen. 5mM Arg is enough to restore the growth of the ∆argB but not ∆argJ in the iron-depleted media.
Minor comments:
Line 42: confusing sentence, not all AA in this list are aromatic
Line 76: full stop is missing
Line 77: “previous studies” as authors refer to two other papers
Line 80: “those studies”
Check the article for “a nitrogen source”